# ARCHITECTURE COMPRESSION

## ABSTRACT

In this paper we propose a novel approach to model compression termed Architecture Compression. Instead of operating on the weight or filter space of the network like classical model compression methods, our approach operates on the architecture space. A 1-D CNN encoder/decoder is trained to learn a mapping from discrete architecture space to a continuous embedding and back. Additionally, this embedding is jointly trained to regress accuracy and parameter count in order to incorporate information about the architecture's effectiveness on the dataset. During the compression phase, we first encode the network and then perform gradient descent in continuous space to optimize a compression objective function that maximizes accuracy and minimizes parameter count. The final continuous feature is then mapped to a discrete architecture using the decoder. We demonstrate the merits of this approach on visual recognition tasks such as CIFAR-10/100, FMNIST and SVHN and achieve a greater than 20x compression on CIFAR-10.

## 1 INTRODUCTION

The recent adoption of CNNs for real-time, on-device applications has fueled a great demand for better model compression. Conventional methods such as quantization, pruning and distillation have proven to be reliable approaches in reducing redundancies in networks. However, one avenue where there has been little progress is that of architecture space based compression.

Approaches such as Iandola et al. (2016), Howard et al. (2017), Rastegari et al. (2016), have introduced hand defined heuristics to design networks that are efficient without sacrificing accuracy. However, designing such networks still remains a laborious task, requiring much human expertise. In such a context, an efficient dataset-driven approach to determine the optimal architecture is desirable.

Recent methods such as Ashok et al. (2018); He et al.; Zhou et al. (2018); Tan et al. (2018) have proposed dataset-driven architecture based model compression using reinforcement learning. However, one drawback of such RL based methods is that they are less sample-efficient, often requiring a large number of architecture evaluations and several hours or days of training to find a single compressed model. Furthermore, it is unclear how these approaches compare to random search and whether they only succeed due to large-scale exploration and evaluation of many architectures.

To the best of our knowledge, this is the first paper to introduce a novel gradient descent based approach to perform architecture compression. To facilitate learning, we describe a mapping from discrete architecture space to a continuous space that encodes the structure of architecture for a specific dataset. We train a one-dimensional convolutional encoder/decoder neural network to learn the structure of discrete architecture space while using accuracy and parameter regressors to impose the structure inherent to learning the dataset. For compression, we encode an input architecture and leverage this learned continuous latent space to perform gradient descent on the encoded feature. The augmented feature is then decoded into a compressed discrete architecture.

We demonstrate the effectiveness of our approach on several visual learning tasks of varying difficulty (FMNIST, SVHN, CIFAR-10, CIFAR-100) and show 5-20x compression on architectures. We also compare this approach to conventional compression methods such as pruning, distillation and reinforcement learning and show that our architectures are smaller and faster than those produced by the baseline methods.

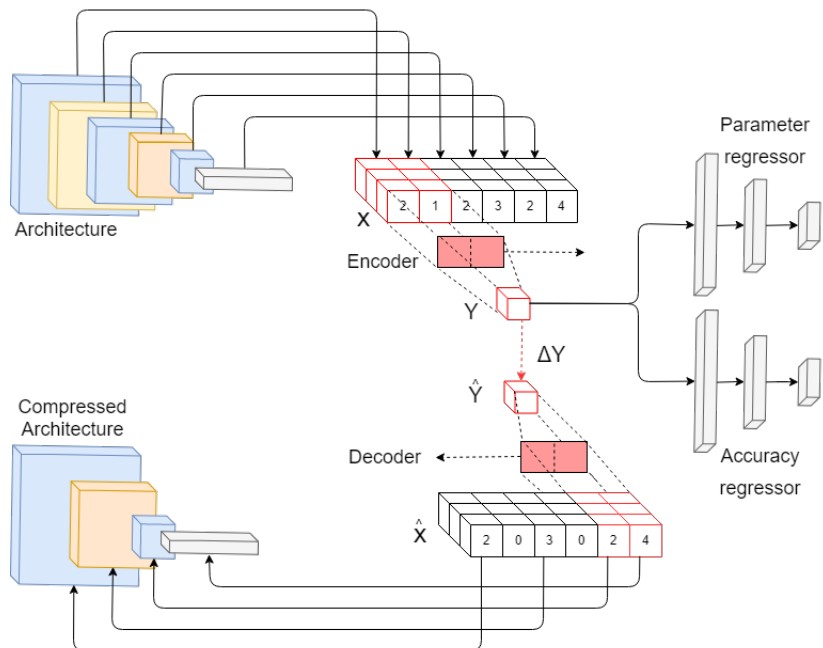

Figure 1: High level overview of 1D-CNN encoder-decoder system

## 2 RELATED WORKS

### 2.1 ARCHITECTURE SEARCH

There have been recent works in architecture search such as Zoph & Le (2016), Baker et al. (2016), Miikkulainen et al. (2017), Real et al. (2017), that use reinforcement learning or evolutionary algorithms to traverse the space of architectures. These approaches characterize architectures as discrete entities in a non-differentiable space and use either a constrained state space or heuristics to discover architectures. These approaches have been subject to criticism from the machine learning community for using large amounts of computation.

Work such as Ashok et al. (2018), Tan et al. (2018), Zhou et al. (2018), He et al. attempt to search for *efficient* architectures instead of just the best. However these approaches also currently rely on reinforcement learning. In contrast to these approaches, our method shows that we can leverage classical optimization techniques such as gradient descent to effectively search the space of architectures for a constrained objective. Methods such as Liu et al. (2018) and Luo et al. (2018) attempt to make the space of architectures differentiable however this work is focused on discovering repeatable convolutional cells instead of optimizing the architecture as a whole.

### 2.2 PRUNING, QUANTIZATION, HAND DESIGNED MODELS

Pruning-based methods preserve the weights that matter most and remove the redundant weights LeCun et al. (1989), Hassibi et al. (1993), Srinivas & Babu (2015), Han et al. (2015b), Han et al. (2015a), Mariet & Sra (2015), Anwar et al. (2015), Guo et al. (2016). While pruning-based approaches typically operate on weight space, our approach operates on the the model architecture. Additionally, our method offers greater flexibility as we can use memory, inference time, power, and other hardware constraints to guide the compression process, resulting in the optimal architecture for a given dataset and constraints.

Hand designed models such as Iandola et al. (2016) and Howard et al. (2017) have been shown to work well in practical applications such as self-driving cars and mobile phones. These are good stepping stones in making progress in designing efficient architectures. Our work focuses on designing a general method to design such architectures in a fully automated data-driven manner.

## 3 STRUCTURE OF ARCHITECTURE COMPRESSION SPACE

In this section we show how to map certain classes of architectures to a unique vector representation in order to enable learning. We then analyze the cardinality of architecture space for compression and prove that this space has several desirable properties for learning a mapping.

### 3.1 TOPOLOGICAL REPRESENTATION OF ARCHITECTURES

We observe that most modern neural network architectures $A \in \mathbb{A}$, with the exception of recurrent neural networks can be expressed as a directed acyclic graph (DAG). We use the property that every DAG has a topological ordering (Proof in section 7) to represent an architecture as a partially ordered sequence of layers, $l_i \in \mathbb{L}$. Formally,

$$l_{1:N} = [l_0, l_1...l_N] \text{ such that } \forall i, j \in [1, N], i > j, \nexists l_i \to l_j \tag{1}$$

where the $\to$ operator denotes that the activation of $l_i$ is the input of $l_j$. Furthermore, this topological ordering is unique (Proof in section 7.2) for standard convolutional networks and those with simple skip connections (e.g. ResNet). We will denote this ordered representation of an architecture as

$$o : \mathbb{A} \to \mathbb{L}^N : A \mapsto l_{1:N} \tag{2}$$

where $o$ is the topological sorting function. Note also that for unique, one-to-one mappings, there exists an inverse function $o^{-1}$ to recover $A$ given $l_{1:N}$

$$o^{-1} : \mathbb{L}^N \to \mathbb{A} : o(A) \mapsto A \tag{3}$$

### 3.2 CARDINALITY OF ARCHITECTURE COMPRESSION SPACE

Let $A \in \mathbb{A}$ be an architecture that is parameterized by $\theta$ and trained on some dataset $D$. Then we can model the accuracy $a \in [0, 1]$ as a continuous random variable stochastically sampled from a distribution $P_\theta(a|A, D)$, since $\theta$ is typically optimized using a variant of stochastic gradient descent.

The expected accuracy $E_\theta[a|A, D]$, while not exactly computable due to the large dimension of $\theta$, can be approximated by training and evaluating the network under varying intializations $\theta_0$, i.e.

$$E_\theta[a|A, D] \approx \frac{1}{N} \sum_i^N g_\theta(A, D, \theta_i) \tag{4}$$

where $g$ minimizes an objective function over $D$ with respect to $\theta$. Thus, we are interested in learning a mapping from discrete architecture space to the expected accuracy

$$f : \mathbb{A} \to [0, 1] \subset \mathbb{R} : A \mapsto E_\theta(a|A, D) \tag{5}$$

However, since we are interested in *model compression*, we also want to construct the inverse mapping from the expected accuracy to discrete architecture space for a given parameter count $p \in \mathbb{R}^+$

$$f_p^{-1} : \mathbb{R}^+ \times ([0, 1] \subset \mathbb{R}) \to \mathbb{A} : E_\theta(a|A, D) \mapsto A \tag{6}$$

In the following subsections, we prove that such a mapping, while not unique, is feasible to learn since the range of valid discrete architectures is finite.

For the following sections, we will denote parametric layers i.e. convolutional layers, batch normalization and fully connected layers as $f_\theta(x)$, where $x$ is the activation of the previous layer and $\theta$ represents the parameters of the layer.

Non-parametric pooling layers as max pooling and average pooling layers are written as $q_s(x)$, where $s > 1$ (no upsampling) is the stride of the layer. Lastly activations such as sigmoid, ReLU, softmax etc. are represented as $\sigma(x)$.

We also make the following practical assumptions in the following proofs:

1. The input $X \in \mathbb{R}^{H \times W \times C}$, where $H, W, C \in \mathbb{N}^+$ and $H, W, C$ are fixed.
2. For layers $l_i \in A$, if $l_i \in \Sigma$, $l_{i+1} \notin \Sigma$, where $\Sigma$ is the set of possible activation functions.

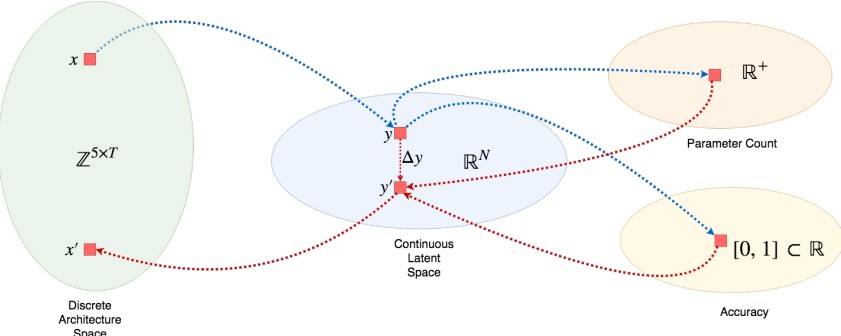

Figure 2: Structure of architecture compression spaces with forward (blue) and backward (red) mappings visualized

3. Pooling layers reduce dimensions of input image by some factor

**Lemma 3.1.** *For a spatial input of dimension $d_0$, the number of pooling layers possible in a valid architecture is finite.*

*Proof.* Let $x_i$ be the input of the ith layer with spatial dimension $d_i \in \mathbb{N}^+$. We observe that a pooling operation with stride $s \in \mathbb{N}^+ > 1$ and kernel size $k \in \mathbb{N}^+$ $x_{i+1} = q_s(x_i)$ results in an output of $d_{i+1} = \lfloor \frac{d_i - k}{s} \rfloor + 1 < d_i$. Since $d_0$ is fixed (Assumption 1) and $s$ is discrete, for any $d_0$ there is a finite number of pooling layers possible in a valid architecture. □

**Theorem 3.2.** *For a given parameter constraint $p$, the set of valid architectures that meet the parameter constraint $A_p = \{A|\ numParams(A) \leq p\}$ is finite.*

*Proof.* Next, suppose we have a partially constructed network with parameter count $p_0$. Then by assumption 2, we can add at most one activation function. Furthermore by Lemma 3.2, we have a finite choice of pooling layers such that the architecture still remains valid.

Suppose we add a parametric layer $l_i = f_{\theta i}$, the parameter constraint exceeded if $p_0 + |\theta_i| > p$. Thus, we have the upperbound $|\theta_i| \leq p - p_0$. Since $|\theta_i|$ is discrete, we have a finite number of choices of layers such that the partially constructed network is still valid.

We then observe that since $d_i$ monotonically decreases if we add a pooling layer and the upperbound of $|\theta_i|$ monotonically decreases if we add a parametric layer.

Since at least every other layer has to be one of these types, the length of any valid architecture is finite. Thus, we arrive at the conclusion that there are a finite number of valid architectures for a given parameter constraint. □

## 4 APPROACH

Having proved that a mapping is feasible, we now describe our approach to train a 1-dimensional convolutional encoder/decoder, an accuracy regressor and a parameter regressor to learn such a mapping (Outline in Fig. 1).

The variable length input vector to the network denoted by $x_{1:T} = [x_i, ..., x_T] \in \mathbb{Z}^{5 \times T}$ is a topologically ordered representation of the architecture as per Section 3.1. Each layer is represented by the hyperparameters of each layer. Specifically, it is a discrete 5 dimensional vector as follows:

$$x_i = [t, k, o, s, p] \in \mathbb{Z}^5 \tag{7}$$

where $t$ is the layer type, $k$ the kernel size, $o$ the number of output channels, $s$ the stride and $p$ the padding.

## 4.1 ENCODER AND DECODER

The encoder is a one-dimensional convolutional neural network that takes as input the discrete architecture representation and outputs a continuous feature. Prior approaches use recurrent networks such as LSTMs or bidirectional RNNs where backpropagating through large sequences can result in vanishing gradients and difficult credit assignment Hochreiter et al. (2001).

In our approach, we use a 1D-CNN encoder for several reasons. It can operate on variable length sequences, CNNs with sufficient depth have a large receptive field over the input sequence and it does not suffer from vanishing gradients since the length of gradient propagation is no longer a function of the sequence length but a function of the depth of the inference network.

The encoder $E$, computes the function:

$$y = E(x_{1:T}) \tag{8}$$

where $x_{1:T} \in \mathbb{Z}^{5 \times T}$ is a variable length sequence representing the architecture and $y \in \mathbb{R}^N$ a continuous feature embedding. Details of the network can be found in the appendix.

The decoder has a similar residual 1D-CNN architecture as the encoder except that convolutions are replaced with transposed convolutions.

The decoder $D$, computes the function:

$$x_{1:T}^* = D(y) \tag{9}$$

where $x_{1:T}^* \in \mathbb{R}^{5 \times T}$ is a continuous variable length sequence representing the architecture and $y \in \mathbb{R}^N$ a continuous feature embedding. The continuous variable $x_{1:T}^*$ is discretized to $\hat{x}_{1:T} \in \mathbb{Z}^{5 \times T}$.

## 4.2 ACCURACY AND PARAMETER REGRESSORS

The accuracy regressor and parameter regressor are two fully connected networks that take the continuous embedding of the architecture $y$ as input and predict the expected accuracy, $E_\theta[a]$ and parameter count, $p$ of the architecture $A$.

The accuracy regressor learns the function

$$a = f_a(y) \tag{10}$$

where $a \in [0,1] \subset \mathbb{R}$ is trained to predict $a^* = E_\theta[a]$ (which is approximated as per 3.2 by training the network with multiple intitializations). This network consists of 3 fc-relu blocks with a sigmoid output activation to convert the output to $[0,1]$.

The parameter regressor learns the function

$$p = f_p(y) \tag{11}$$

where $p \in \mathbb{R}^+$ is trained to match the ground truth parameter count $p^*$. The ground truth parameter count is computed as $p^* = \frac{\sum_i^N |l_i| - \bar{p}}{\sigma_p}$ where $|l_i|$ is the number of parameters of each layer in the network. The mean parameter count $\bar{p}$ is subtracted to center the parameter count around 0 and the standard deviation of the parameter count $\sigma_p$, is used to scale the parameter count for better conditioning of the range of the output. This network consists of 3 fc-relu blocks with a ReLU function to predict a non-negative real scalar.

## 4.3 OPTIMIZATION

In order to learn to encode/decode the architecture and predict accuracy/parameter count, we jointly optimize a multi-task loss function. While all four tasks are formulated as regression problems, each has a differing domain. To account for this, we choose appropriate loss functions to best suit each task.

The decoder is trained to approximate positive integral values of varying scale since we may have small values for padding, kernel size, type and stride and large values for number of outputs. To enable faster convergence, we use a mean squared error.

$$L_d(\hat{x}_{1:T}, x_{1:T}) = \frac{1}{T} \sum_{i=0}^{T} |\hat{x}_i - x_i|_2 \tag{12}$$

A standard L-1 loss is used for the accuracy regressor since the output of the sigmoid function is bounded between $[0, 1]$.

$$L_a(\hat{a}, a^*) = |\hat{a} - a^*|_1 \tag{13}$$

For training the parameter regressor, we use the Huber loss with $\delta = 1$. This loss is less sensitive to outliers in the data and does not suffer from exploding gradients as described in Girshick (2015). This is important since the domain of the parameter count is $\mathbb{R}$ and it is possible that the training set may have outliers such as large networks that do not converge.

$$L_p(\hat{p}, p^*) = \begin{cases} \frac{1}{2}(\hat{p} - p^*)^2 & \text{if} |\hat{p} - p^*| \leq 1 \\ |\hat{p} - p^*| - \frac{1}{2} & \text{otherwise} \end{cases} \tag{14}$$

The network is then trained to minimize the joint loss function:

$$L = L_d + \lambda_1 L_a + \lambda_2 L_p \tag{15}$$

### 4.4 ARCHITECTURE COMPRESSION USING GRADIENT DESCENT

---
**Algorithm 1** Architecture Compression

---
1: **procedure** TRAIN$(\mathcal{A}, D)$
2:     **for** $\mathcal{A}$ in Architectures **do**
3:         $a^* \leftarrow \text{train}(A, D)$
4:         $p^* \leftarrow (\text{num\_params}(A) - \bar{p})/\sigma_p$
5:         $x \leftarrow o(\mathcal{A})$
6:         $y \leftarrow \text{encoder}_{\theta_1}(x)$
7:         $\hat{a} \leftarrow \text{acc\_regressor}_{\theta_1}(y)$
8:         $\hat{p} \leftarrow \text{param\_regressor}_{\theta_3}(y)$
9:         $\hat{x} \leftarrow \text{decoder}_{\theta_4}(y)$
10:        $L = L_d(\hat{x}, x) + \lambda_1 L_a(\hat{a}, a^*) + \lambda_2 L_p(\hat{p}, p^*)$
11:        $\theta = \text{argmin}_\theta L$
12:     **end for**
13: **end procedure**
14: **procedure** COMPRESS$(\mathcal{A})$
15:     $x \leftarrow o(\mathcal{A})$
16:     $y \leftarrow \text{encoder}_{\theta_1}(x)$
17:     $\hat{a} \leftarrow \text{acc\_regressor}_{\theta_2}(y)$
18:     $\hat{p} \leftarrow \text{param\_regressor}_{\theta_3}(y)$
19:     $L_c = L_a(\hat{a}, 1) + \lambda L_p(\hat{p}, 0)$
20:     $y \leftarrow y + \eta \frac{dL_c}{dy}$
21:     $x_{\text{comp}} \leftarrow \text{decoder}_{\theta_4}(y)$
22:     $\mathcal{A}_{comp} \leftarrow o^{-1}(x_{\text{comp}})$
23: **end procedure**

---

Once the inference network has converged and can predict the accuracy and number of parameters of a network, we use the network for architecture compression.

In order to compress an architecture, we perform gradient descent on the continuous embedding space using the compression objective function.

$$L_c(\hat{a}, \hat{p}) = L_a(\hat{a}, 1) + \lambda L_p(\hat{p}, 0) \tag{16}$$

The compression objective function simultaneously minimizes the number of parameters and maximizes the accuracy of the network to achieve compression. We can also incorporate known hardware constraints $p_0$ or accuracy requirements $a_0$ to relax this objective function by modifying it as follows

$$L_c(\hat{a}, \hat{p}) = L_a(\hat{a}, a_0) + \lambda L_p(\hat{p}, p_0) \tag{17}$$

The gradient from this objective is then backpropagated to the continuous embedding $y$ with learning rate $\eta$ to generate a new embedding $y'$.

$$y' = y + \eta \frac{dL_c}{dy} \tag{18}$$

The decoder converts this embedding into a temporal sequence $x_{\text{comp}}$ that is then discretized and converted into an architecture $A_{\text{comp}} = o^{-1}(x_{\text{comp}})$.

# 5 EXPERIMENTS

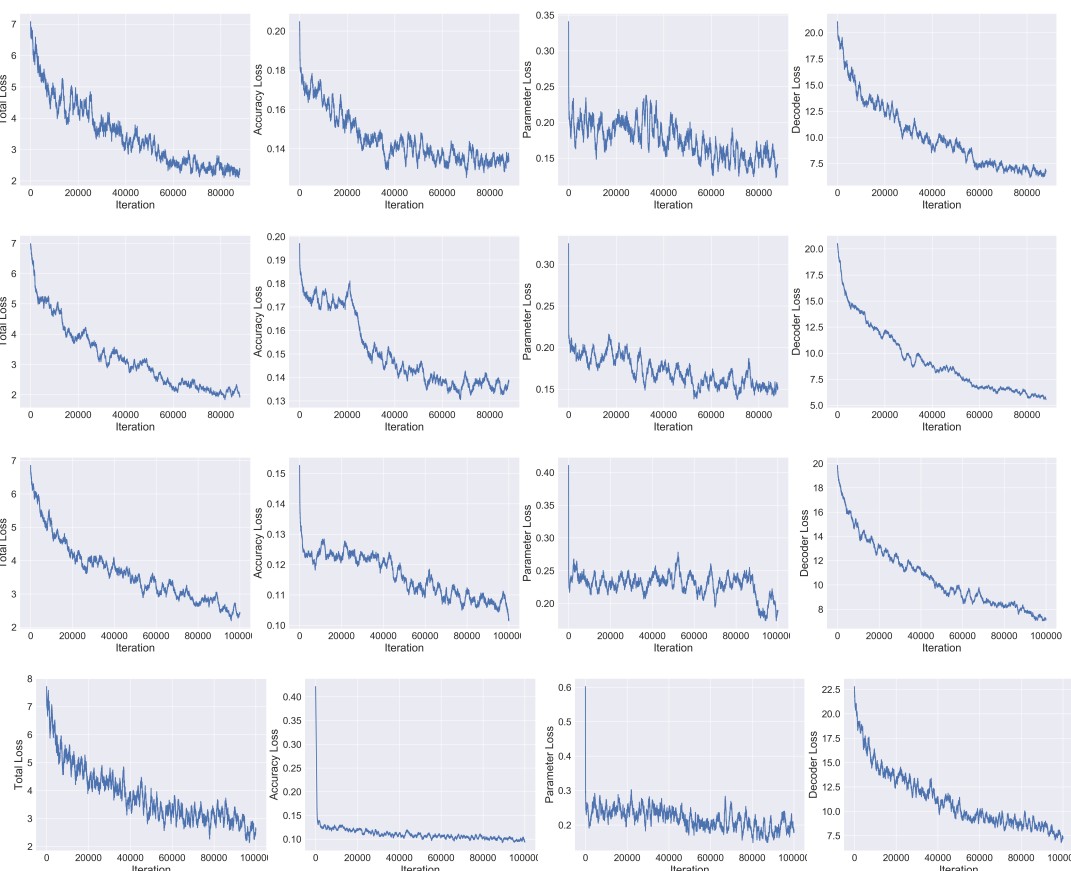

Figure 3: Joint training loss on (from top-bottom) **FMNIST**, **SVHN**, **CIFAR-10**, **CIFAR-100**

In this section, we empirically evaluate our approach, showing that it is capable of compressing networks by upwards of 10x by benchmarking them on modern classification tasks such as CIFAR-10, CIFAR-100, FMNIST and SVHN.We show that our approach outperforms current state of the art architecture compression techniques by comparing the results to several baselines.

In the following experiments, we used a randomly generated dataset of 1500 architectures for training the encoder/decoder and regressors. Each architecture was trained for each task for 5 epochs with m=5 different random initializations to obtain a target expected accuracy. As observed in Ashok et al. (2018); Zoph & Le (2016); Tan et al. (2018), 5 epochs of training seems to provide sufficient signal about the convergence characteristics of the network. All of the following experiments were run on 2x NVIDIA 1080TI GPUs. Additional training details such as choice of optimizer and hyperparameters are included in Section 10 in the appendix.

## 5.1 DATASETS

**FMNIST** The Fashion-MNIST Xiao et al. (2017) dataset consists of $28 \times 28$ pixel grey-scale images depicting images of fashion products from 10 categories. We use the standard 60,000 training images and 10,000 test images for experiments.

**CIFAR-10** The CIFAR-10 Krizhevsky & Hinton (2009) dataset consists of 10 classes of objects and is divided into 50,000 train and 10,000 test images (32x32 pixels). This dataset provides an

incremental level of difficulty over the FMNIST dataset, using multi-channel inputs to perform model compression.

**SVHN** The Street View House Numbers Netzer et al. (2011) dataset contains 32x32 colored digit images with 73257 digits for training, 26032 digits for testing. This dataset is slightly larger that CIFAR-10 and allows us to observe the performance on a wider breadth of visual tasks.

**CIFAR-100** To further test the robustness of our approach, we evaluated it on the CIFAR-100 dataset. CIFAR-100 is a harder dataset with 100 classes instead of 10, but the same amount of data, 50,000 train and 10,000 test images (32x32). Since there is less data per class, there is a steeper size-accuracy tradeoff.

## 5.2 COMPRESSION EXPERIMENTS

Table 1: Summary of compression results

| Model | Acc. | #Params | $\Delta$ Acc. | Compr |
|-------|------|---------|---------|-------|
| Fashion-MNIST | | | | |
| CONV-7 | 91.46% | 708K | +0.98% | 7.61x |
| CIFAR-10 | | | | |
| CONV-10 | 92.35% | 24.4M | -0.04% | 20.33x |
| CIFAR-100 | | | | |
| CONV-10 | 70.95% | 24.4M | -1.32% | 4.51x |
| SVHN | | | | |
| CONV-10 | 96.02% | 24.4M | -0.63% | 8.76x |

In this section we evaluate the ability of our approach to compress architectures. For our experiments, we use a standard convolutional architecture consisting of stacked conv-bn-relu blocks and a fully connected layer. We use a 7 block network CONV-7 for Fashion-MNIST and 10 block network CONV-10 for the others. Table shows original average accuracy of the model on the validation set, followed by the number of parameters in the original model followed by two columns for the improvement in accuracy in the compressed architecture and the compression rate. We notice a minor improvement in accuracy by the compressed architecture in Fashion-MNIST while observing a minor drop in the other datasets. Additionally, our method is able to achieve solid compression on all the datasets and up to 20x compression on CIFAR-10 (1.2M parameters for final compressed architecture).

### 5.2.1 BASELINES

We compare our approach to various model compression baselines including reinforcement learning, pruning and knowledge distillation on the CIFAR-10 dataset. The experiments show that our approach is able to find a smaller and more accurate model than the other approaches.

Table 2: Baselines on CIFAR-10

| Model | Acc. | #Params |
|-------|------|---------|
| Romero et al. (2014) (Distillation) | 91.33% | 1.2M |
| Molchanov et al. (2016) (Pruning) | 91.06% | 2.3M |
| Ashok et al. (2018) (RL) | 92.05% | 1.7M |
| **Ours** | **92.31%** | **1.2M** |

## 6 CONCLUSION

In conclusion, we have described a novel approach to compress CNNs by first training a continuous embedding on a representation of the architecture and then performing gradient descent to determine an optimal architecture for the given task. We also introduced a novel theoretical analysis of CNNs which we hope will inspire future work. We also demonstrate that our method performs well over a variety of computer vision datasets. Given that this is a novel direction of research, we note that there exist multiple future directions to go. Expanding the search space to include networks of greater complexity such as ResNets or DenseNets would be of practical interest. Analyzing the transfer learning properties of this approach via the reuse of the weights for other tasks would be of great practical use as well.

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

## 7 APPENDIX

### 7.1 TOPOLOGICAL ORDERING

**Lemma 7.1.** *If G is a directed acyclic graph (DAG), then G has a topological ordering*

*Proof.* We prove this by induction on $n$, where $n = |G|$.

**Base case:**

For $n = 1$, the statement is true since the topological ordering is $G$.

**Hypothesis:**

Assume that there exists a topological ordering on all DAGs $G$ of size $k$.

**Inductive step:**

Given a DAG $G'$ with $k + 1$ nodes, select a node $v$ with no incoming edges.

Then $G* = G' - \{v\}$ is a DAG since deleting $v$ cannot induce a cycle on $G'$.

Since $|G*| = k$, the hypothesis implies that $G*$ has a topological ordering $T*$.

Since $v$ has no incoming edges, no partial order is violated by placing $v$ at the head of the topologically ordered sequence.

Thus $T = [v, T*]$ is a valid topological ordering of $G'$ and the lemma is proven. $\square$

**Lemma 7.2.** *If a topological sort $l_{1:T} = [l_0...l_i...l_T]$ has the property that $\forall i, \exists\, edge(i, i+1)$, then it is unique*

*Proof.* It follows directly that the above property implies that there exists a Hamiltonian path on the graph since traversing the graph in the topologically sorted order satisfies each node being visited once. We can then show that the existence of a Hamiltonian path in a DAG implies unique ordering.

Suppose a DAG has two Hamiltonian paths, then let $x \in P_1, y \in P_2$ be the first nodes on the paths $(P_1, P_2)$ that differ.

This implies that there is a path from $x \to y$ that is a subpath of $P_1$ and a path from $y \to x$ that is a subpath of $P_2$. This implies a cycle in the graph, which is a contradiction. $\square$

## 7.2 VISUALIZATION OF 1D-CNN FILTERS

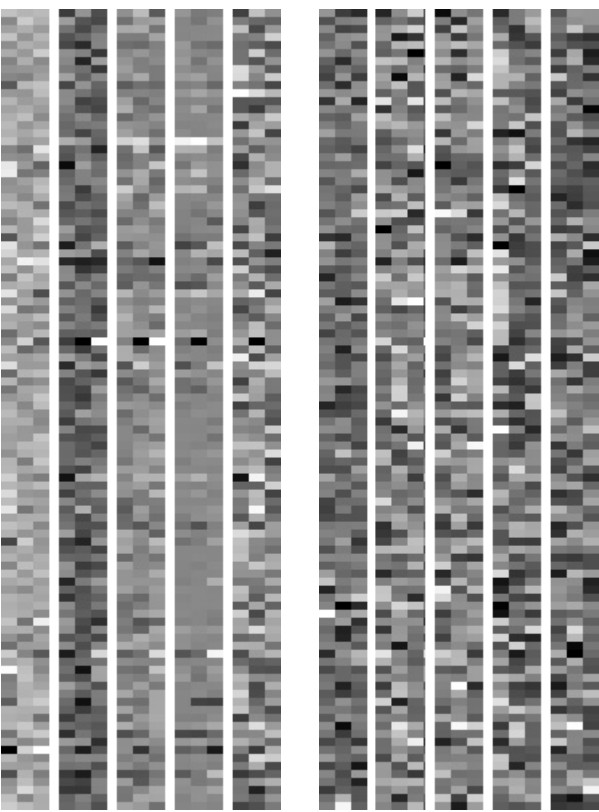

Figure 4: First 100 layer 1 filters for encoder (left) and decoder (right). Each column represents an input channel and rows represent filters.

## 8 DETAILS ON 1-D CNN ARCHITECTURE

Let $x_i \in \mathbb{Z}^{K \times T}$ be the ith activation or in the case of the first layer, the discrete representation of an architecture. Then a 1-d convolution operation applies a filter $w \in \mathbb{R}^{L \times K}$ to a window of $L$ features in the input vector. We then add a bias $b \in \mathbb{R}$ to this output, to produce a feature map $h_i$. Finally we apply a batch normalization function $g$, an activation function $\sigma$ and add the residual to produce a

new activation $x_{i+1}$ as follows. Appropriate padding is added to the output in order to preserve the dimension and allow element-wise addition of the residual.

$$h_i = w * x_i + b$$
$$\hat{h}_i = g(h_i)$$
$$x_{i+1} = \sigma(\hat{h}_i) + x_i$$

In the case of a strided convolution where the temporal dimension of the output is smaller than the original input, we apply a convolution with the same stride $s$. The operations then become the following.

$$h_i = f_s(x_i, w_1, b_1)$$
$$\hat{h}_i = g(h_i)$$
$$\hat{x}_i = f_s(x_i, w_2, b_2)$$
$$x_{i+1} = \sigma(\hat{h}_i) + \hat{x}_i$$

where $f_s(x, w, b)$ performs convolutions on input $x$ with stride $s$, weights $w$ and bias $b$.

## 9    SAMPLING OF RANDOM ARCHITECTURES

In section 4.4, we describe a set of randomly generated architectures that are trained and then used to learn a continuous architecture search space. In this section, we detail how this set of architectures is generated.

In order to generate a sensible set of architectures, we randomly sample architectures and then reject degenerate architectures.

First we sample the length of the architecture randomly from a discrete unifrom distribution $l \sim U(2, 50)$. We set the lower bound to be 2 since we need at least one classification layer (fully connected) and one convolution layer for the architecture to be valid. Furthermore, we set the upperbound to 50 layers as an arbitrary limit that is within experimental constraints.

After this, we sample the 5 layer configuration variables for each of the $l$ layers randomly. Specifically, $t \sim U(1, 8)$ where:

1. Convolution
2. MaxPool
3. BatchNorm
4. ReLU
5. Sigmoid
6. Dropout
7. AveragePool
8. Linear

Additionally, $k \sim U(1, 10)$, $s \sim U(1, 10)$, $o \sim U(16, 4096)$, $p \sim U(1, 10)$.

For layers that do not have certain configuration variables (e.g. batch normalization has no stride), we set the variable to be 0. For certain layers like MaxPool, ReLU or BatchNorm that do not change the number of output channels, we set the variable to the number of input channels. Furthermore, We also enforce that the number of input channels to the first is the same as the input of the task example and that the final layer is a fully connected layer with the number of outputs corresponding to the number of classes in the task.

Lastly, we filter out degenerate architectures if they do not contain any convolutional layers, the output dimension is too small to proceed or the memory/computational footprint is too large to be practical.

## 10 TRAINING DETAILS

### 10.1 RANDOMLY GENERATED ARCHITECTURES

This section describes the implementation details for the training procedure of the randomly generated architectures. All the experiments used the Adam optimizer and were run on the PyTorch framework. The same procedure was used to train the final output architecture. **Fashion-MNIST** The architectures for FMNIST were trained for 50 epochs with a starting learning rate of 0.01. The learning rate is reduced by a factor of 10 in the 30th epoch. A batch size of 64 was used.
**CIFAR-10/100** The architectures for CIFAR-10/100 were trained for 150 epochs with a starting learning rate of 0.001. The learning rate is decreased by a factor of 10 in the 80th and 120th epochs. Standard data augmentation with horizontal mirroring (p=0.5), random cropping with padding of 4 pixels and mean subtraction of (0.5, 0.5, 0.5). A batch size of 128 was used.
**SVHN** The architectures for SVHN were trained for 150 epochs with a starting learning rate of 0.001. The learning rate is decreased by a factor of 10 in the 80th and 120th epochs. Mean subtraction of (0.5, 0.5, 0.5) and a batch size of 128 was used.

### 10.2 ARCHITECTURE COMPRESSION SEARCH

The encoder, decoder and regressors were optimized using the SGD optimizer with nesterov momentum=0.5 and a learning rate of 0.003. The CIFAR-10/100 networks were trained jointly for 100000 iterations while the SVHN and FMNIST ones were trained for 80000 iterations. Furthermore, learning rate decay was used with period 30 and decay multiplier 1/5.

