# OpenReview forum: "Architecture Compression"
_ICLR.cc/2019/Conference_

### Official Review · AnonReviewer3 · 2018-11-01
**Interesting underlying idea, but evaluation insufficient**

**Rating:** 4
**Confidence:** 4

**Review:**

This paper deals with Architecture Compression, where the authors seem to learn a mapping from a discrete architecture space which includes various 1D convnets. The aim is to learn a continuous latent space, and an encoder and decoder to map both directions between the two architecture spaces. Two further regressors are trained to map from the continuous latent space to accuracy, and parameter count. By jointly training all these networks, the authors are now able to compress a given network by mapping it's discrete architecture into the latent space, then performing gradient descent towards higher accuracy and lower parameter count (according to the learned regressors).

The authors perform experiments on 4 standard datasets, and show that they can in some cases get a 20x reduction in parameters with negligible performance decrease. They show better Cifar10 results than a few baselines - I am not aware whether this is SOTA for that parameter budget, and the authors do not specify.

Overall I really like the idea in this paper, the latent space is well justified, but I cannot recommend acceptance of the current manuscript. There are many notational issues which I go into below, but the key issue is experiments and reproducability.

The search space is not clearly defined. Current literature shows that the performance of these methods depends a lot on the search space. The manuscript does make clear that a T-layer CNN is represented as a 5XT tensor, with each column representing layer type, kernel size etc. However the connectivity is not defined at all, which implies that layers are simply sequentially stacked. This seems to preclude even basic architectural advancement like skip connections / ResNet - the authors even mention this in section 3.1, and point to experiments on resnets in section 4.4, but the words "skip" and "resnet" do not appear anywhere else in the paper. I presume from the emphasis on topological sort that this is possible, but I don't see how.

If this paper is simply dealing with linear chains of modules, then the mapping to a continuous representation, and accuracy regression etc would still be interesting in principle. However it does mean that essentially all the big architecture advancements post-VGG (ie inception, resnet, densenet...) are impossible to represent in this space. Most of the Architecture Search works cited do have a search space which allows the more recent advances.

I don't see a big reason why the method could not be extended - taking the 5D per-layer representation and adding a few more dimensions to denote connectivity would seem reasonable. If not, the authors should clearly mention the limitations of their search space.


In terms of experiments, Figure 3 is very hard to interpret. The axes labellings are nearly too small to read, but it's also unclear what loss this even is - I presume this is the 'train' loss of L_d + \lambda_1L_a + \lambda_2L_p, but it could also be the 'compress' loss. It also behaves very unusually - the lines all end up lower than where they started, but oscillate around a lot, making me wonder if the curves from a second set of runs would look anything alike. It's not obvious why there's not just a 'normal' monotonic decrease.

A key point that is not really addressed is how well the continuous latent space actually captures what it should. I am extremely interested to know whether the result of 'compress', ie a new concrete architecture found by gradient descent in the latent space, actually has the number of parameters and the accuracy that the regressors predict. This could be added as columns in Table 1 - eg the concrete architecture for Cifar10 gets 20.33x compression and no change in accuracy, but does the regressor for the latents space predict this compression ratio / accuracy as well? If this is the case, then I feel that the latent space is clearly very informative, but it's not obvious here.

It would also be really useful to see some concrete input / output values in discrete architecture space. Presumably along the way to 20x compression of parameter count, the optimisation passes through a number of progressively smaller discrete architectures - what do these looks like? Is it progressively fewer layers / smaller filters / ??? Given that the discrete architecture encoding appears to have a fixed length of T, it's not even clear how layers would be removed. Figure 1 implies you would fill columns with zeros to delete layers, but I don't see this mentioned elsewhere in the text.

More minor points:

Equation numbers would be extremely useful throughout the paper.

Notation in section 3 is unclear. If theta represents trained parameters, then surely the accuracy on a given dataset would be a deterministic value. Assuming that the distribution P_{\theta}(a | A, D) is used to represent the non-determinism of SGD training, is \theta supposed to represent the initialised values of the weights?

There are 3 functions denoted by 'g' defined on page 3 and they all refer to completely different things - this is unnecessarily confusing.

The formula for expected accuracy - surely this should be averaging over N different training / evaluation runs, something like:

E_{\theta}[a | A, D] \simto \frac{1}{N} \sigma_{i}^N g_{\theta}(A, D, \theta_i)

The decoder computes a 6xT output instead of a 5xT output - what is this extra row for?

In the definition of "ground truth parameter count" p^* - presumably the standard deviation here is the standard deviation of the l vector? This formulation is a bit surprising, as convolutional layers will generally have few parameters, and final dense layers could have many. Did you consider alternative formulations like simply taking the log of the number of parameters? Having a huber loss with scale 1 for this part of the loss function was also surprising, it would be good to have some justification for this (ie, what range are the p^* values in for typical networks?)

In algorithm 1 line 4 - here you are subtracting \bar{p} from num_params before dividing by standard deviation, which does not appear in the formulation above.

In the experiments:
How were the 1500 random architectures generated? I presume by sampling uniformly a lot of 5xT tensors, but this encoding is not clearly defined. x_i is defined as being in the set of integers, does this include negative numbers? What are the upper / lower limits, and is there anything to push towards standard kernel sizes like 3x3, 5x5, etc? These random architectures were then trained five times for 5 epochs - what optimizer / hyperparameters / regularization was used? Similarly, the optimization algorithm used in the outer loop to learn the {en,de}coders/regressors is not specified.

I would move the lemma and theorem into the appendix - they seem quite unrelated to the overall thrust of the paper. To me, saying that an embedding is not uniquely defined, but can be learnt is not that controversial, and I don't need proofs that some architecture search space has a finite number of entries. Surely the fact that the architecture is represented as a 5xT tensor, and practically there are upper limits to kernel size, stride etc beyond which an increase has no effect, already implies a finite space? Either way, this section of the paper did not add much value from my perspective.


I want to close by encouraging the authors to resubmit after addressing the above issues, I do believe the underlying idea here is potentially very interesting.

---

> ### Author Response · Authors · 2018-11-24
> **Thank you for your detailed and insightful comments! (Part 2)**
>
>
> >> There are 3 functions denoted by 'g' defined on page 3 and they all refer to completely different things - this is unnecessarily confusing.
> >> The formula for expected accuracy - surely this should be averaging over N different training / evaluation runs
> >> The decoder computes a 6xT output instead of a 5xT output - what is this extra row for?
>
> ---
> We have updated the paper to take these suggestions into account and changed the notation on page 3 such that o - topological ordering function, q - pooling layers and g - loss minimization function. We hope this makes the notation clearer.
>
> >> In the definition of "ground truth parameter count" p^* - presumably the standard deviation here is the standard deviation of the l vector? This formulation is a bit surprising, as convolutional layers will generally have few parameters, and final dense layers could have many.
>
> ---
> The parameter count is predicted for the entire network and thus the standard deviation is that of the entire network, not per each layer.
>
> >> Did you consider alternative formulations like simply taking the log of the number of parameters? Having a huber loss with scale 1 for this part of the loss function was also surprising, it would be good to have some justification for this (ie, what range are the p^* values in for typical networks?)
>
> ---
> We did try taking the log of the parameters but found that the precision of the parameter regressor decreased. We think this is related to the spread of parameter counts we see in the dataset and that smaller networks might have noisier predictions as a result of the log-scaling. We found empirically that a scale of 1 worked well. p^* values range from about approximately [-10, 10] for our dataset.
>
> >> In algorithm 1 line 4 - here you are subtracting \bar{p} from num_params before dividing by standard deviation, which does not appear in the formulation above.
>
> ---
> In our experiments we subtract the mean to center the parameter count, we have updated the formulation to reflect this.
>
> >> How were the 1500 random architectures generated?
> >> These random architectures were then trained five times for 5 epochs - what optimizer / hyperparameters / regularization was used?
> >> Similarly, the optimization algorithm used in the outer loop to learn the {en,de}coders/regressors is not specified.
>
> ---
> We have included training details for both the architectures as well as the compressor networks in the appendix along with details for the random architecture generation.
>
> >> Surely the fact that the architecture is represented as a 5xT tensor, and practically there are upper limits to kernel size, stride etc beyond which an increase has no effect, already implies a finite space?
>
> ---
> In this paper, we are concerned with the cardinality of the set of networks since we desire to form a mapping from architecture space to some latent space. Without a parameter constraint, there could theoretically be an infinite number of valid networks.

---

> ### Author Response · Authors · 2018-11-24
> **Thank you for your detailed and insightful comments! (Part 1)**
>
>
> We would like to thank the reviewer for their detailed review and numerous useful comments, we have addressed the comments and incorporated the changes below.
>
> >> This seems to preclude even basic architectural advancement like skip connections / ResNet - the authors even mention this in section 3.1, and point to experiments on resnets in section 4.4, but the words "skip" and "resnet" do not appear anywhere else in the paper. I presume from the emphasis on topological sort that this is possible, but I don't see how.
>
> ---
> Our original plan was indeed to add two more variables specifying the position of the layer relative to the start and end of the skip connection in the input. However, this would require more manual tuning to ensure that the dimension of the residual is the same as that of the output at the end of the skip connection. We have updated that particular sentence to make things clearer.
>
> Given that this is a novel direction of research, we have chosen to start off by demonstrating results on simpler models before modifying the search space to suit custom models. We do think that incorporating a wider selection of networks is a direction indeed worth expanding upon. We have added a note about this in the conclusion.
>
> >> In terms of experiments, Figure 3 is very hard to interpret. The axes labellings are nearly too small to read, but it's also unclear what loss this even is - I presume this is the 'train' loss of L_d + \lambda_1L_a + \lambda_2L_p, but it could also be the 'compress' loss.
>
> ---
> We have updated the figures to be easier to read as well as re-run the experiments with updated hyperparameters to generate plots for each of the losses separately so that it is clearer as to which objective is being minimized. We do also incorporate optimization techniques such as alternating the loss that is optimized during each iteration which could explain the non-monotonic decrease.
>
> >> A key point that is not really addressed is how well the continuous latent space actually captures what it should. I am extremely interested to know whether the result of 'compress', ie a new concrete architecture found by gradient descent in the latent space, actually has the number of parameters and the accuracy that the regressors predict.
>
> ---
> The predicted accuracy and compression are accurate when the updated feature in latent space is close to the original. We found that after several iterations (~20), we start to see divergence. The degree of accuracy varies for each query network. An analysis will be included in the appendix.
>
> >> It would also be really useful to see some concrete input / output values in discrete architecture space. Presumably along the way to 20x compression of parameter count, the optimisation passes through a number of progressively smaller discrete architectures - what do these looks like?
>
> ---
> With a larger step size or aggressive parameter count reduction, we observe that the number of filters decreases, more layers become closer to identity (more consecutive ReLUs, MaxPool with kernel size 1) and fewer conv layers are produced. Interestingly, when ==accuracy is maximized, the network also learns to produce higher capacity networks. An analysis will be included in the appendix.
>
>
> >> Given that the discrete architecture encoding appears to have a fixed length of T, it's not even clear how layers would be removed. Figure 1 implies you would fill columns with zeros to delete layers, but I don't see this mentioned elsewhere in the text.
>
> ---
> We naturally remove the layer when the number of filters, stride or kernel size is 0. We have updated the paper to make this clearer
>
>
> >> Equation numbers would be extremely useful throughout the paper.
>
> --
> We have updated the paper with equation numbers for better readability.
>
> >> Notation in section 3 is unclear. If theta represents trained parameters, then surely the accuracy on a given dataset would be a deterministic value. Assuming that the distribution P_{\theta}(a | A, D) is used to represent the non-determinism of SGD training, is \theta supposed to represent the initialized values of the weights?
>
> ---
> In our formulation, \theta is a random variable representing the trained parameters where the randomness originates from the random initialization procedure and the SGD training. It is true that accuracy is deterministic given a specific sample was drawn from the distribution, but it can vary across two samples.

---

### Official Review · AnonReviewer2 · 2018-11-03
**Interesting idea but the approach is not rigorous**

**Rating:** 4
**Confidence:** 3

**Review:**

The paper presents a way to compress the neural network architecture. In particular, it first extracts some characteristics for the neural network architecture and then learns two mapping functions, one from the encoded architecture characteristics to the expected accuracy and the other from the same encoded architecture characteristics to the number of parameters. In the meanwhile, the proposed approach learns the encoding and the decoding for the architecture characteristics.

Pros:
1. The idea of converting the architecture characteristics, which is discrete in nature, to continuous variables is interesting. The continuity of the architecture characteristics can help architecture search tasks.

Cons:
1. My main concern is the validity of the compression step, Procedure COMPRESS, in Algorithm 1. First, is only one step gradient descent applied? If it is, why not minimize the L_c until convergence?  Second, it seems that minimizing L_c cannot guarantee that both error and the number of parameters are reduced. It is possible that only one of them is reduced.
2. The writing of the paper needs to be improved. Some notations are not consistent with each other. For example, the loss notations in Line 19 in Algorithm 1 are different from those defined in Sec. 4.3.
3. There is no step size, \eta in Line 20 in Algorithm 1, but there is a step size in the last equation on Page 6.
4. It is unclear to me how the hyperparameters, such as the step size and \lambda's, are chosen.
5. More experimental results are needed to support the proposed approach.

In summary, I think this paper is not ready to be published.

 ==== After rebuttal ====
The authors' feedback clarified some of my concerns. But my main concern about why minimizing the objective function can reduce both error and the number of parameters still remains. So I changed my rating to 4 from 3.

---

> ### Author Response · Authors · 2018-11-24
> **Thanks for the comments and questions about the paper!**
>
> >> My main concern is the validity of the compression step, Procedure COMPRESS, in Algorithm 1. First, is only one step gradient descent applied? If it is, why not minimize the L_c until convergence?
>
> ----
> We repeat the gradient descent procedure while the true compression of the model improves and the true accuracy does not decrease too much. In practice, we observed that testing models every 5 steps (with sgd, momentum=0.5, lr=0.003) works well and fewer than 10 iterations are necessary.
>
> >> Second, it seems that minimizing L_c cannot guarantee that both error and the number of parameters are reduced. It is possible that only one of them is reduced.
>
> ---
> We weight the accuracy objective slightly lower than the parameter count objective and find that this helps to maintain accuracy while minimizing parameter count. The weight value is a hyperparameter that varies for each dataset and model.
>
> >> The writing of the paper needs to be improved. Some notations are not consistent with each other. For example, the loss notations in Line 19 in Algorithm 1 are different from those defined in Sec. 4.3.
> >> There is no step size, \eta in Line 20 in Algorithm 1, but there is a step size in the last equation on Page 6.
>
> ---
> We have updated the notation to match those defined in section 4.3
>
>
> >> It is unclear to me how the hyperparameters, such as the step size and \lambda's, are chosen.
> The optimal hyperparameters differ for each dataset and network. While there are a few approaches to choosing hyperparameters such as Bayesian optimization or grid search, we found that in practice, the loss plots are quite informative in determining which loss to weight more. For example, if the total loss does not seem to decrease but the parameter loss is still high, we increase the weight of the parameter loss.
>
> ---
> We observed that the step size for the compression step is hard to determine a priori but developed a few ways to tune it. We usually start off at a small value (0.003) and run the compression procedure for 5 steps until we arrive at a compressed network.

---

### Official Review · AnonReviewer1 · 2018-11-05
**Good paper, experimental validation must be improved**

**Rating:** 6
**Confidence:** 4

**Review:**

Even though many people have considered prunning as architecture search, it has not been explored enough so far. This paper comprises a good approach for compression of achitectures using pruning. Based on the uniqueness of the topological ordering of commonly used neural networks (feed forward, skip connection), the paper proposes a simple and easily manipulable vector (sequence) representation for a wide class of neural networks. Instead of using RNN networks, such long seqeunce representation are mapped to a continuous embedding by 1D-CNN. While training this embedding, for the purpose of compression, predictors needed for compression are jointly trained with embedding.
Consequently, the proposed method presents a possiblity of including many other constraints during the architecture search.

In the specification of layers, the layer type is just appointed to an integer variable, even though in reality the layer type is a categorical variable. This choice is ok for standard neural network layers, where effectively the choice is between a single catecorical aspect. However, for more sophisticated layer configurations, where you may need many categorical choices, this model choice will not be adequate and will likely lead to artificially biased design choices. The authors should explain the limitations of this model design and propose methods these limitations can be tackled.

The overall model achieves quite good result in compression. On CIFAR10 the model show good performance as compared to existing compression methods. It should be noted that other methods start with a given stucture, so their search space is more limited than this paper's approach. Specifically, compared to those methods, the search space for the proposed paper is larger because although the number of layers is fixed, the connections between layers give more freedom to the compression algorithm.

Currently, the number of experiments is borderline. They are enough to indicate the potentials of this approach. However, additional experiments would be welcome. For one, it would be to evaluate the proposed model in more challenging setups: evaluate on ImageNet dataset, using some of the recent architectures (e.g., ResNet, VGGnet, and so on). What is more, for more compressed architectures with better accuracy, when searching for a compressed architecture global optimization methods like Bayesian Optimization is worth to try, for instance using the recently proposed BOCK (Oh et al, ICML 2018).

Some additional comments.
- For a finite number of edges the number of possible graphs (including the valid architecture) are finite when the input is finite and pooling reduces the feature map size. Thus, it seems that the statement in theorem 3.2 is rather trivial and it is not worth calling it a Theorem.

Overall, this was an interesting paper to read and worth of acceptance, provided that the proposed method delivers also in more competitive experimental settings.

---

> ### Author Response · Authors · 2018-11-24
> **Thank you for your insightful and useful comments!**
>
> We thank you for your insightful and detailed comments on improving the paper. We have addressed the comments and made changes where appropriate.
>
> >> In the specification of layers, the layer type is just appointed to an integer variable, even though in reality the layer type is a categorical variable.
>
> ---
> This is an interesting idea and one that we did consider. The main reason we decided to go with a regression-based approach is that for an arbitrary task, it is unclear how many categories we would need to determine the appropriate capacity (filter size, number of filters). Under the categorical formulation, we would need to redefine the learning problem for every different task, thus complicating transfer learning. Another approach that was considered was to structure the decoder as a classification problem for variables that were categorical and regression for the other variables. However since the simpler regression approach worked, we decided to stick with that approach.
>
> >> For one, it would be to evaluate the proposed model in more challenging setups: evaluate on ImageNet dataset, using some of the recent architectures (e.g., ResNet, VGGnet, and so on).
>
> ---
> While the number of architectures that have to be trained (1500 trained for 5 epochs) is relatively small compared to other architecture search papers, training on large datasets like ImageNet pose a challenge due to the number of computing resources needed to run the experiment.
> Regarding recent architectures, we note that the conv architectures used in the paper are very similar to the VGG architecture and achieve better performance with fewer parameters. Using more complex architectures such as ResNets or DenseNets is an interesting future direction which would require several architecture-specific modifications.
>
> >> For a finite number of edges the number of possible graphs (including the valid architecture) are finite when the input is finite and pooling reduces the feature map size. Thus, it seems that the statement in theorem 3.2 is rather trivial and it is not worth calling it a Theorem.
>
> ---
> Even if there exists pooling that reduces feature map size, it is possible to have an infinite number of graphs if there is no parameter constraint. An example of this is as follows: consider a network with finite input I. Let the network be I-> MaxPool -> Conv1….->ConvN, where conv preserves feature dimensions. A new graph with a finite number of edges can be formed by adding ConvN+1. Thus, there can be an infinite number of possible graphs each having a finite size.

---

> > ### Comment · AnonReviewer1 · 2018-12-10
> > **Thank you for your response**
> >
> > Thank you for the response.
> >
> > I agree with the concerns raised by the other reviewer on the experiment including reproducibility and multi-target optimization.  Still, I think the point in the rebuttal is not critical enough to change my decision. That said, I do believe as long as there is enough pooling layer (which reduces feature map size), calling the statement 3.2 a theorem still looks like overstating.

---

### Meta-Review · Area_Chair1 · 2018-12-14
**Interesting idea, but requires additional experimentation and analyses**

**Confidence:** 4
**Recommendation:** Reject

**Metareview:**

The authors propose a scheme to learn a mapping between the discrete space of network architectures into a continuous embedding, and from the continuous embedding back into the space of network architectures. During the training phase, the models regress the number of parameters, and expected accuracy given the continuous embedding. Once trained, the model can be used for compression by first embedding the network structure and then performing gradient descent to maximize accuracy by minimizing the number of parameters. The optimized representation can then be mapped back into the discrete architecture space.
Overall, the main idea of this work is very interesting, and the experiments show that the method has some promise. However, as was noted by the reviewers, the paper could be significantly strengthened by performing additional experiments and analyses. As such, the AC agrees with the reviewers that the paper in its present form is not suitable for acceptance, but the authors are encouraged to revise and resubmit this work to a future venue.